# Impact of Parental Knowledge and Beliefs on HPV Vaccine Hesitancy in Kenya—Findings and Implications

**DOI:** 10.3390/vaccines10081185

**Published:** 2022-07-26

**Authors:** Chester O. Kolek, Sylvia A. Opanga, Faith Okalebo, Alfred Birichi, Amanj Kurdi, Brian Godman, Johanna C. Meyer

**Affiliations:** 1Migori County Referral Hospital, Suna 40400, Kenya; chester.kolek@students.uonbi.ac.ke; 2Department of Pharmacy, University of Nairobi, Nairobi 00202, Kenya; faith.okalebo@uonbi.ac.ke; 3Directorate of Pharmaceutical Services, Kenyatta National Hospital, Nairobi 00202, Kenya; abirichi@knh.or.ke; 4Department of Pharmacoepidemiology, Strathclyde Institute of Pharmacy and Biomedical Sciences, University of Strathclyde, Glasgow G4 0RE, UK; amanj.baker@strath.co.uk; 5Department of Pharmacology, College of Pharmacy, Hawler Medical University, Erbil 44001, Iraq; 6Department of Public Health Pharmacy and Management, Sefako Makgatho Health Sciences University, Ga-Rankuwa, Pretoria 0208, South Africa; hannelie.meyer@smu.ac.za; 7Centre of Medical and Bio-Allied Health Sciences Research, Ajman University, Ajman 13306, United Arab Emirates

**Keywords:** willingness, hesitancy, knowledge, beliefs, HPV vaccination, parents, Kenya

## Abstract

Cervical cancer can be prevented by human papillomavirus (HPV) vaccination. However, parents can have concerns about vaccinating their daughters. Consequently, there is a need to identify prevalence and risk factors for HPV vaccine hesitancy among parents in Kenya. A descriptive cross-sectional study was conducted among parents with children aged 9–14 years attending a leading referral hospital in Kenya. Data on sociodemographic traits, HPV knowledge, beliefs and vaccine hesitancy were collected. Out of 195 participants, 183 (93.5%) were aged >30 years. Thirty-four (46.4%) of males and 39 (35.1%) of females did not know that the vaccine is given to prevent HPV infection. Encouragingly, levels of vaccine acceptance were high (90%) although one-third (37.9%) had a negative perception about the effectiveness of the vaccine, with vaccine hesitancy attributed to safety concerns (76%) and feelings that the child was too young (48%). Positive beliefs and knowledge of the vaccine were positively associated with parental willingness to vaccinate their children. Low levels of parenteral education and a younger age among mothers were negatively associated with willingness to vaccinate. Most parents (59%) would consult their daughters before vaccination, and 77% (*n* = 150) recommended early sex education. Despite low knowledge levels, there was high parental willingness to have their children vaccinated.

## 1. Introduction

Cancer is the leading cause of death among women, especially in low-income countries [1], with cervical cancer recording the highest mortality among all cancers of the reproductive system [2,3]. In 2020, globally, there were 604,127 new cervical cancer cases annually, with 341,831 deaths in the same year [4]. Whilst prevalence and mortality rates from cervical cancer have reduced with the introduction of the human papillomavirus (HPV) vaccine as primary prevention of cervical cancer, coupled with early detection and other strategies, the World Health Organization (WHO) estimated that in 2019 globally over one million women were still living with cervical cancer, with 300,000 women dying unnecessarily each year from cervical cancer, especially in low- and middle-income countries (LMICs) [2,5,6,7]. Consequently, the WHO still considers cervical cancer among the greatest threats to the health of women worldwide. In view of this, the goal of the WHO and others is to continue to appreciably reduce current prevalence and mortality rates [4]. This is especially important in LMICs, which currently account for approximately 85% of preventable deaths from cervical cancer each year [4,5,8,9].

In LMICs, cervical cancer ranks fourth among all female cancers in terms of morbidity after breast cancer, colorectal, lung cancer in that order (estimates for 2020), with a ratio of mortality to incidence of 57% [3]. In addition, LMICs have the highest mortality rates globally from cervical cancer, accounting for nearly 90% [1,10]. East African nations have among the highest cancer mortalities worldwide [11,12]. Kenya currently has a high cervical cancer incidence, with an estimated age-standardized incidence rate of 31.3 per 100,000 women per year in 2020 [3]. These estimates are among the highest cervical cancer morbidity and mortality rates globally [12,13]. The high cervical cancer burden across LMICs has been attributed to several factors. These include attitudinal and knowledge barriers, inadequate health infrastructure and systems as well as a lack of screening and treatment programs [5,6,10,14]. Having said this, multiple activities over the years in Rwanda have resulted in Rwanda having a high vaccination rate among girls to help reduce future morbidity and mortality [5,13,15]. The multiple activities in Rwanda include pre-immunization efforts to manage logistics, nurses and community health workers spending considerable time promoting the HPV vaccine for cancer prevention and dispelling myths, as well as government officials, spiritual leaders, teachers and volunteers supporting educational initiatives to enhance vaccination rates along with advertising [5,13,16].

Among the leading risk factors for cervical cancer is prolonged exposure to HPV [2]. Early sexual debuts, unprotected sex, and multiple sexual partners also increase the probability of developing cervical cancer [17,18,19]. Of all the sexually transmitted diseases, HPV has the highest incidence [18]. It is estimated that three-quarters of all sexually active people have had this virus at one point in their lives and that approximately 35% of women get exposed to the virus within the first two years of their sexual debut [20].

There is also a link between HIV infection and higher rates of HPV acquisition [21,22]. This is because of a decreased clearance rate of HPV and precancerous lesions leading to an elevated risk of cervical cancer. Women with low CD4 counts and high viral loads have an increased risk of HPV infection [23], with low CD4 counts causing decreased HPV clearance. Consequently, LMICs with higher HIV risks such as Sub-Saharan African countries tend to have a higher prevalence of cervical cancer, unless pro-active measures are undertaken as seen in Rwanda [15,24].

Published studies have shown that the HPV vaccine is safe and effective against cervical cancer [25,26,27]. The vaccine achieves maximal prophylaxis against precancerous cervical changes upon administration before sexual debuts in the target adolescent population [11,27]. As a result, successful vaccination campaigns can help achieve the WHO target of eliminating cervical cancer by 2030 [28]. However, successful immunization against HPV requires most persons eligible to be immunized [10].

Following HPV vaccine development, the Global Alliance for Vaccines and Immunization (GAVI) supported vaccine demonstration projects in 2012 among LMICs. GAVI supported and co-funded 2-year HPV vaccine demonstration projects with a view to eventually planning and implementing nationwide HPV vaccination programs among countries in Sub-Saharan Africa as seen in Rwanda. Kenya received GAVI support [27], with the Kenyan HPV vaccine 2-year (2013–2015) demonstration project undertaken in February 2014 in Kitui county [13,20]. A school-based approach was used, which achieved a coverage of 96%; however, it was resource intensive [20]. The HPV vaccine was eventually introduced into the national routine immunization schedule in Kenya in October 2019 [29]. The program targeted girls of 10 years of age, prior to their likely sexual debut, to be executed in schools and facilities in partnership with the various regions in Kenya as the needs dictated. This is facilitated by the fact that the HPV vaccine in Kenya is currently made freely available via donors. However, coverage in Kenya in 2020 for the 1st dose was only 33% and the second dose only 16% [30]. This needs to be urgently addressed given current high cervical cancer rates in Kenya [3,30].

However, there can be concerns towards the HPV vaccine across countries, exacerbated by misinformation, impacting on its uptake [30,31,32,33,34]. The WHO refers to this as hesitancy “a delay in acceptance or refusal of vaccination despite availability of vaccination services” [35]. Parents’ moral responsibility towards their adolescent’s health in Kenya makes them key determinants regarding whether pre-adolescent girls get vaccinated or not. Whilst we are aware that a number of studies on vaccine hesitancy have been undertaken, most of these have been conducted in higher-income countries [36]. We are also aware that a limited number of studies have been undertaken in Kenya, aimed at assessing parental determinants of vaccine hesitancy including knowledge and attitudes towards the HPV vaccine [20,24,37]. When undertaken, these studies have tended to be performed among females. However, males potentially transmit the virus and they are key household decision makers in the Kenyan context. Consequently, they are indirectly affected by cervical cancer. Knowledge, as well as attitudes concerning the vaccine, impact on uptake rates that further affect successful immunization. We are aware of studies assessing attitudes and knowledge of HPV vaccine among teachers in Kenya as well as studies assessing potential methods to vaccinate hard-to-reach populations in Kenya [20,38]. We are also aware of a similar previous study conducted in Kitui county [20]. However, we are unaware of studies that had assessed vaccine hesitancy rates among parents in Kenya, especially males, in recent years. This is important due to the current high cervical cancer rates in Kenya coupled with low vaccination rates exacerbated by misinformation [30,39]. Consequently, the aim of this study was to identify the determinants of vaccine hesitancy amongst parents attending clinics in Kenya to inform future policies. Patient co-payments are not an issue with HPV vaccination, unlike in many other disease areas, with vaccines provided free of charge via donor schemes [40,41,42].

## 2. Materials and Methods

### 2.1. Study Design, Location and Population

This was a descriptive cross-sectional study conducted among adults of either sex seeking care at the medical clinics of Kenyatta National Hospital (KNH), between June and August 2020. KNH was selected for this study as it is a leading public tertiary care hospital in the Capital City of Kenya, Nairobi, representing an urban population. KNH is also the largest referral facility in East and Central Africa [43], and most likely to have an appreciable number of potential interviewees. The study population included both male and female Kenyan parents with pre-adolescent and adolescent children, attending KNH medical outpatient clinics. We are aware that we did not include parents from rural areas of Kenya with issues of education and accessibility likely to impact on potential findings [30]. However, there was lockdown and other measures during the COVID-19 pandemic in Kenya that could adversely affect recruitment [44,45]. Consequently, for this initial study, we concentrated on KNH.

### 2.2. Sample Size Determination

Sample size determination was calculated using the Cochran formula [46], as this study was descriptive and we were of potential issues surrounding larger sample sizes [47]. A systematic review found that the prevalence of HPV vaccine hesitancy amongst adults ranged from 15 to 33% [48]. Assuming a vaccine hesitancy prevalence of 15%, a minimum sample size of 195 was calculated, at a 95% confidence level and with a 5% margin of error.

### 2.3. Eligibility Criteria, Sampling and Participant Recruitment

The patient numbers attending the clinics during the COVID-19 pandemic reduced substantially; despite this, the numbers attending the clinic were large enough to achieve the desired sample size. The hospital had restricted attendance during the earlier days of the pandemic, but these restrictions were already lifted at the time of this study. Participants were eligible if they were aged above 18 years and were a parent to a pre- or adolescent child aged 9–14 years. In addition, they had to be able to communicate fluently either in Swahili or English, visited the medical clinic of KNH during the study period and provided informed written consent for participation. Women and men who self-reported that they had been previously diagnosed and treated for either cancer of the cervix, penile cancer or genital warts, were excluded from this study. These parents were excluded as by experiencing the disease, they would have better knowledge of cervical cancer and might be more receptive regarding the vaccine, leading to biased responses. As a result, further compromise the representativeness of the study findings.

The hospital has eight outpatient pharmacies, each serving specific types of patients. One of them (Pharmacy 15) serves the medical clinics, which target patients with chronic comorbidities such as hypertension and diabetes. Medical clinics are held daily, each booking approximately 30 patients per day. Pharmacy 15 was identified as most suitable for recruitment of participants. This is because patients typically collect their medication from this pharmacy after consultation at the medical clinics. Most of the patients attending medical clinics are aged above 40 and therefore more likely to have adolescent children. Participants were approached and recruited as they collected their medicines from Pharmacy 15.

We aimed to recruit 10 patients daily as it is a realistic number of participants that could be handled comfortably by the data collectors. Since approximately 30 patients are seen daily, a quasi-randomized sampling method was used. Hence, every third patient was approached when they presented themselves to collect their medication, aiming at the desired sample size of 10 participants per day. In the queue, participants were briefly informed about this study and invited to participate. Those who expressed interest in participating were invited to the Medicines Information Centre office, which is a private room located in the pharmacy for further screening and potential recruitment. They were subsequently screened more thoroughly for eligibility, and written informed consent was obtained for those willing to participate. On some days, the targeted sample size of 10 per day was not achieved because some of the invited potential participants did not meet the eligibility criteria or declined to participate during the screening process. Sampling and recruitment continued on a daily basis, until the desired sample size was achieved.

### 2.4. Research Instruments and Data Collection

Data were collected using a questionnaire administered in a face-to-face interview by the data collectors. The questionnaire was adopted from previous studies [20,49], and subsequently modified to suit the objectives of this study. Modifications included additional questions about parents’ willingness to consult their child before making a decision about getting the HPV vaccine. The questionnaire collected information on sociodemographic characteristics, knowledge and beliefs towards HPV infection and acceptance of the HPV vaccine. There were also questions regarding religious beliefs as this has influenced vaccine uptake rates in Kenya [39]. Additionally, patients were asked if they would consider their child’s preferences with regard to getting vaccinated. Responses to questions on attitude towards the HPV vaccine were in the form of a 5-point Likert scale. The other questions were closed-ended and required simple yes and no responses. The questionnaire is available in Appendix A.

The questionnaire was subsequently pretested on 20 study participants who met the eligibility criteria before the principal study was rolled out. The findings showed the instrument’s internal consistency was sufficient as indicated by a Cronbach’s alpha value of 0.78. Questions where there were concerns with comprehension were subsequently modified before commencement of actual data collection.

### 2.5. Data Management

Within 24 h of data collection, responses were entered into an Epi Info^TM^ version 7.2.4.0 (Centers for Disease Control and Prevention, Atlanta, GA, USA) database. The data were cleaned to ensure consistency in the format of the responses. Missing entries were identified, and an attempt was made to identify sources of error in data collection and rectify them as far as possible to ensure data completeness. Only the principal investigator had access to the documents, with the database password protected and backed-up daily to minimize accidental data loss. The database was locked at the end of data collection so that any future fraudulent entries could be tracked. All data are archived for 10 years in line with Kenyan law on archiving of research data.

### 2.6. Variables and Data Analysis

We used three outcome variables, namely the score obtained on the knowledge regarding HPV infection and the HPV vaccine, beliefs about HPV vaccine score and their willingness (acceptance) to have their child vaccinated. The first two outcome variables were numeric and the last was dichotomous. The key predictor variable was parental sex. The other predictor variables included sociodemographic traits and specific aspects of knowledge.

The knowledge score for each individual was obtained by calculating the sum of all the correct responses to the 30 questions that were designed to elicit information on knowledge on HPV infection, cervical cancer and the HPV vaccine. The maximum score was 30. The scores were subsequently converted to a dichotomous variable (0 for No or ‘I don’t know’ and 1 for Yes). A cut off value of greater than 15 for the 30 questions on knowledge was used to show greater knowledge, based on prior research.

The belief score was obtained by the simple sum of the responses to the questions on attitudes and beliefs. The responses were categorized as negative beliefs (1), not sure (2), and positive beliefs (3). The maximum score was nine.

Categorical variables were summarized as frequencies and percentages. The continuous variables were tested for normal distribution using the Shapiro–Wilk test and histograms were plotted to examine the distribution. Normally distributed variables were summarized as the mean with a standard deviation, and those not normally distributed were summarized as the median and interquartile range.

Inferential analysis was conducted to compare the characteristics of parents who were willing to have their children vaccinated with those who were unwilling. Inferential tests used were the unpaired t-test for continuous variables, the Mann–Whitney test for variables not normally distributed and the chi square test for categorical variables.

Because the knowledge and belief scores were not normally distributed and therefore generalized, additive regression modelling was used to identify the key determinants of knowledge.

Logistic regression was undertaken to identify factors that influence parental willingness to vaccinate their children. Multivariate regression controlled for confounding such as education levels. Covariates that were significant on bivariate inferential analysis were included in the regression models. The parsimonious models were identified using stepwise backward elimination. The level of significance was set at 0.05. Data cleaned in Epi Info were exported to STATA Version 13 and R 4.3 software for statistical analysis. The *mgcv* package in R was used for general additive modelling.

### 2.7. Ethical Considerations

Approval to conduct this study was obtained from Kenyatta National Hospital—University of Nairobi (KNH/UoN) research and ethics review committee (Reference number: KNH-ERC/A/178). The Kenyatta National Hospital administration and Head of Department of the medical clinics granted permission to conduct this study. Voluntary informed consent was sought from all study participants before they started the questionnaire. Confidentiality and privacy were maintained by replacing identifier information with a code and database protection.

## 3. Results

### 3.1. Recruitment and Baseline Sociodemographic Characteristics of Participants

We approached a total of 488 people. Out of these, 147 (30%) declined to participate in this study. The remainder, 341 (70%), were screened for eligibility, of whom 195 (57%) met the inclusion criteria and agreed to participate. Most parents recruited into this study were aged above 30 years (93.5%), with more females than males and most being self-employed. Secondary education was the highest level of attainment for most participants (87; 44.6%). Nearly all participants were married (165; 84.6%) and the majority of participants had children aged between 12 and 14 years. Other than three participants, all were Christians.

We did not collect data on income since the HPV vaccine is provided free of charge through donor programs. More details on the baseline sociodemographic characteristics of the parents are summarized in Table 1.

### 3.2. Overall Knowledge Score

The histogram summarizing the performance of the participants, with a maximum value of 30, is presented in Figure 1.

Figure 1 illustrated that the knowledge score was not normally distributed and skewed to the right (median: 16; IQR: 17 to 21). Lack of normal distribution was confirmed using the Shapiro–Wilk test (*p* < 0.001). Data transformations were applied to make the variable conform to the normal distribution; however, these transformations were not successful.

### 3.3. Sources of Information and Knowledge of HPV Infection, Cervical Cancer and Its Prevention

The most common source of information on the HPV vaccine was the participants’ fellow workers (82% females and 72% males), while religious leaders were the least common source of information. Women had a greater variety of information sources compared to men, with more females obtaining their information from health care providers (*p* = 0.025). The findings on the sources of information on the HPV vaccine are summarized in Appendix A.

Table 2 presents the proportion of correct responses to selected questions regarding knowledge of the HPV vaccine across males and females. Females generally had more knowledge about the HPV vaccine compared to males, with a number of statistically significant differences between the knowledge of males and females. A summary of the differences for specific aspects of HPV vaccination is presented in Table 2.

More females (92; 82.9%) compared to males (61; 72.6%) knew that the HPV vaccine was available free of charge for adolescent girls. In addition, 71.2% of participating females knew the vaccine was for the prevention of cervical cancer (*p* = 0.040). Of the female participants, 64.9% (72) accurately stated that the vaccine is for the prevention of HPV infection. For both males and females, less than 36% knew that the vaccine is useful for the prevention of genital warts; in this aspect, males performed better than females. More women (75.5%) than men (57.1%) knew that young adults could also be vaccinated (*p* = 0.021). Of concern is that there were misleading beliefs with 40.1% of participants falsely thinking there was no need for a Pap smear after getting the vaccine. Worryingly, many participants could also not link the virus to cervical cancer and only slightly more than 50% had heard of HPV infection. Overall, 60% of women were aware that HPV is linked to cervical cancer as opposed to only 41.7% of male participants (*p* = 0.035). Generally, there was poor knowledge about the mode of transmission, risk factors, and symptoms, with less than 40% of participants who responded correctly to these questions.

Generally, there was good knowledge about cervical cancer with almost all participants having heard about cervical cancer (95%). With regard to knowledge on cervical cancer and HPV infection, there were no statistically significant differences in the numbers of correct responses among males and females. Unfortunately, though, few male parents knew that cervical cancer could be diagnosed using the Pap smear test (85% females vs. 36.9% males; *p* < 0.001). Encouragingly, approximately 80% of participants knew that cervical cancer is preventable.

Appendix A provide additional information on the proportions of males and females who answered correctly regarding the various aspects of cervical cancer and knowledge about HPV infection.

### 3.4. Determinants of Knowledge of HPV Infection, Cervical Cancer and HPV Vaccination

Determinants of the knowledge score were identified using bivariate and multivariate generalized additive modelling, with covariates being the sociodemographic characteristics of the participants (Table 3).

With the bivariate analysis, there was no variable that showed a statistically significant correlation with the knowledge score. In order to understand the nature of the interaction, a box plot was generated. This showed that the males’ level of knowledge was highest amongst younger respondents and the knowledge score was lower among older males.

On the other hand, amongst females, the knowledge score was lowest amongst younger females but increased with increasing age. With the bivariate analysis, there was no association between education level and the sex of the respondents.

### 3.5. Parental Views on Knowledge Empowerment and Beliefs about the Safety and Effectiveness of the HPV Vaccine

More females (65%) than males believed that the vaccine is effective; however, there were no statistically significant differences across the sexes. More women (71%) than men (64%) considered the vaccine to be safe (*p* = 0.041). Both sexes (77%) would recommend the vaccine to a 10-year-old.

Although more than 60% of participants had positive beliefs about the safety and efficacy of the HPV vaccine, very few were confident about the information they have to make an informed decision to vaccinate their daughters. Less than 33% were confident they had enough information to make a decision to vaccinate their daughters, with more than 90% of participants expressing an interest in knowing more about the HPV vaccine.

There was a positive attitude towards empowering adolescent children with knowledge on reproductive health. More than 75% of parents recommended that 10-year-old girls should be provided with sex education. Appendix A provide additional information on the respondents’ beliefs and the desire for knowledge empowerment.

### 3.6. Determinants of Beliefs about the HPV Vaccine

The influence of sociodemographic characteristics and knowledge on beliefs about the HPV vaccine is presented in Table 4. The belief score was calculated as the sum of the responses to three questions on beliefs.

With the bivariate analysis, only two variables were associated with the belief score. In the multivariate analysis, only one association remained statistically significant. There was a positive association between occupation and beliefs. Self-employed individuals (not in formal employment) had the highest score (adjusted beta coefficient 0.256; 95% Cl: −0.001, 0.512).

Knowledge was the strongest determinant of the belief score. There was a negative association between having adequate knowledge and beliefs (adjusted beta coefficient −0.06; 95% Cl: 0.039, 0.081). This means that increasing knowledge about HPV may not necessarily translate into positive beliefs. Education status had no association with the belief score.

### 3.7. Parental Willingness to Have Their Children Vaccinated and Reasons for Vaccine Hesitancy

Both males and females showed a high level of willingness (90%) to vaccinate their children. Reasons given for the willingness to have their child vaccinated included positive peer pressure (52%); request by the adolescent (60%); government and school’s recommendation (68%); and doctor’s recommendations (86%). Overall, 94% of the participating female parents were willing to accept the HPV vaccine because of its effectiveness compared to 85% among their male counterparts. Figure 2 summarizes reasons for parental HPV vaccine hesitancy.

More than 70% of parents felt that they did not have adequate information for decision making concerning vaccinating their children despite generally positive attitudes. None of the males were sympathetic about pain at the injection site and less than 26% of parents would reject the vaccine because of their child’s refusal. Encouragingly, less than 4% of the women and 1% of men were against all vaccinations. Similarly, across both males and females, more than 74% would reject the vaccine due to HPV vaccine-related safety concerns if these occurred. Short-term side effects were considered a hindrance by 45% or more of parents. Nearly half the parents felt that their daughters were too young to be vaccinated.

Similarly, 25% or more of parents would reject the vaccine because of concerns that their daughters would be stigmatized due to accusations of being promiscuous after being vaccinated. Of note was that 16% of females and 13% of males would reject the vaccine because their religion does not allow vaccination. In addition, more females than males considered the vaccine to be unnecessary. More females than males would also reject the vaccine for the children if they were to incur any vaccine-related cost such as transportation to the health facility. All reasons identified for acceptance are summarized in Appendix A.

### 3.8. Logistic Regression Analysis for Determinants of Willingness of Parents to Vaccinate Their Child against HPV

Table 5 summarizes the bivariate and multivariate analysis for determinants of willingness of parents to have their child vaccinated. The religion variable could not be used because nearly all participants were Christians.

There was a positive association between willingness to vaccinate their child and beliefs (aOR 2.395; 95% Cl: 1.604, 3.577) and between willingness and knowledge levels (aOR 1.133; 95% Cl: 1.050, 1.222). After adjusting for confounding variables with the knowledge score and the beliefs score, males were more willing than females to have their children vaccinated even though they had less knowledge; however, the measure of association was not statistically significant (aOR 2.369; 95% Cl: 0.911, 7.643). This means that men who had similar levels of knowledge as women would have been more willing to have their children vaccinated.

With regard to sex, qualitative confounding was observed in that the crude measure of association showed a negative association between sex and willingness to have the child vaccinated. After adjusting for confounding, the association increased in magnitude and direction. The crude measure of association was confounded by knowledge.

There was a significant negative association between age and willingness to have their child vaccinated (*p* = 0.040). In addition, there was a negative association between parental education level and willingness to vaccinate. The higher the education level, the lower the willingness to vaccinate their children. Level of occupation and marital status had no association with willingness to vaccinate their children.

## 4. Discussion

We believe this is the first study regarding HPV vaccination in Kenya to include both men and women. This is important because both parents, particularly males, play a crucial role in initiating vaccine visits to clinics in Kenya and previous studies conducted in Kenya did not evaluate sex-specific parental perspectives on willingness to vaccinate their adolescent children.

Overall, males had less knowledge of the HPV vaccine for decision making than their female counterparts, similar to studies conducted in Malaysia [50] and India [51]. This may well be due to a perception among men that HPV and cervical cancer are health matters related to women [52]. The feminization of the HPV vaccine, which has led to female-focused interventions, has resulted in systemic neglect of the male sex in the HPV vaccine campaigns, which is a concern [53]. Consequently, there have been global calls for sex-neutral HPV vaccination initiatives [52,54]. In view of this, media outlets should be increasingly encouraged to portray maternal and reproductive health as an issue that also involves male partners. Men should also be encouraged to actively participate in reproductive health issues since when vaccine advocacy is focused on males, they are typically in favor of protecting women, and therefore, enhance vaccine uptake [55].

Despite low knowledge levels about the HPV vaccine, and a high prevalence of negative beliefs, parental willingness was high with 90% of participating parents willing to have their children vaccinated. Similar findings were noted in a Chinese study that found a lower level of knowledge among junior students but a high willingness to be vaccinated [56]. This represents a social phenomenon where people do receive services which they have little understanding about, and may well reflect the emotive nature of cancer [57,58]. However, blind acceptance can be a concern as seen with high utilization of hydroxychloroquine and ivermectin for patients with COVID-19 despite limited evidence and fears with side effects [59,60,61], as well as acceptance of misinformation regarding HPV vaccines [39]. A high willingness to have the child vaccinated against HPV is positive for future programs in Kenya, similar to Australia, Brazil and Sweden [30,62,63].

Encouragingly as well, parents were willing to involve their adolescent children in the decision-making process. This finding highlights the need to sensitize both parents and adolescent children simultaneously to improve future HPV vaccination rates. Positively, over 77% of the parents favored their children obtaining sex education at an early stage.

Of concern is that approximately 10% of parents were hesitant to have their children vaccinated, with safety concerns (76%) a notable reason. This is similar other studies in Kenya revealing safety concerns [11,17,29,30] alongside inadequate information [20]. Lack of knowledge regarding the HPV vaccine among health care workers, alongside misinformation that the vaccine is a form of contraception and encourages pre-marital sex, are concerns that need to be addressed going forward [29,30,64,65]. Consequently, there needs to be efforts among all key stakeholders to alleviate safety and other concerns, with vaccination rates in Rwanda improved following multiple interventions [16,66]. Having said this, published studies in the United States showed that improving the knowledge levels of parents had no effect on vaccination willingness [67]. However, other studies in the United States have shown a strong positive association between knowledge regarding the HPV vaccine and willingness to have their children vaccinated, similar to studies in Nigeria and Thailand [68,69,70]. On a positive note, 94% of the respondents in our study were interested in obtaining more information, and this desire for more information is an opportunity for increasing HPV vaccine uptake.

A key concern was that the child was too young to be vaccinated with the HPV vaccine, which has been reported in other studies [71], with some parents feeling that the sexual debut of their children was later than the recommended age of HPV vaccination [72]. These further highlight the need for parental education to improve future vaccination rates.

This is important as our study found a negative association between parental level of education with willingness to vaccinate. These negative associations were significant even after adjusting for beliefs and knowledge. Similar findings were seen in Thailand, although this was not statistically significant [69], although contrasting with findings in the United States [73]. Consequently, these finding may be continent specific.

We also found a negative association between older parents and their willingness to have their children vaccinated. This differs though from findings in Thailand and the United States [69,73]. Consequently, both issues should be borne in mind in Kenya when developing educational programs to enhance future vaccination rates.

Most participants in our study had heard about HPV from their colleagues at work. Health officers and television were also mentioned as a significant source of health information. This differs from findings in the United States amongst African Americans which found that medical professionals were the most important source of information for parents, followed by the television [74].

Another concern was the general poor knowledge about HPV, and its association with cervical cancer, and the availability of the HPV vaccine among both male and female participants, similar to a previous study in Kenya study [20] and a systematic review among Sub-Saharan African countries [75]. Interestingly, younger women were found to know less than males of the same age group about HPV. However, older females had a higher score than males of their age group. This may well be due to younger females being disadvantaged due to cultural issues, early school dropout rates and early marriages in Kenya than other countries [76]. Such findings should be borne in mind when developing future educational programs directed at parents in Kenya and other similar countries.

Finally, another key concern was that a significant portion of the study population held incorrect beliefs with only 60% of respondents believing that the vaccine was effective. However, this was higher than among adult women in the USA, where only 29.8% believe that the HPV vaccine effectively prevents cervical cancer [77].

Overall, our findings highlight the need for a multipronged approach to improve future HPV vaccination rates in Kenya, similar to the situation in Rwanda [5,15,16]. This includes reviewing the education of all key health care workers to ensure they are fully conversant with knowledge regarding HPV vaccination to alleviate concerns they may have as well as those of parents and children [30,65]. There also needs to be effective communication campaigns among key influences including church elders in Kenya given their influence [30,39]. This needs to be combined with comprehensive social media campaigns given their increasing influence especially among the younger age group [30,61]. Key messages need to be carefully crafted that address the myths and misconceptions concerning the HPV vaccine to enhance future vaccination rates.

We are aware of several limitations of this study. Firstly, this study was conducted in an urban area where parents were well educated and had better access to the HPV vaccine and information. Consequently, the findings may not be extrapolated to rural settings in Kenya. Secondly, we are aware that the sample was taken from an outpatient clinic and not a general population, which might reflect healthy behavior seekers. In addition, we did not explore the effects of religious dominations and culture on parental decisions as nearly all the participants were Christians. This is important given the influence of religious leaders in Kenya on vaccine usage [30,39].

Cultural aspects, such as limited decision making by women and stigma associated with reproductive issues, were also not adequately explored. Variables affecting vaccine access such as distance to the nearest public facility, knowledge about where the vaccine can be obtained, and the costs of travel to clinics were also not explored. There may also exist provider-related barriers to access to vaccines such as responsiveness, which were not addressed. However, despite these limitations we believe our findings are robust providing direction to the authorities in Kenya on ways to reduce vaccine hesitancy where this exists. We will be following this up in future research projects building on the suggestions arising from our research.

The responses to questions on attitude and beliefs should also have been subjected to item analysis so that the overall score should have been weighted to levels of difficulty. This is an area for future research.

## 5. Conclusions

Despite the low levels of knowledge and safety concerns, there was high parental willingness to have their children vaccinated. The key reason for vaccine hesitancy was insufficient knowledge and safety concerns. Men generally had lower levels of knowledge regarding the HPV vaccine compared to females. Key areas need to be addressed going forward given the low vaccination rates that existed in Kenya in 2020.

## Figures and Tables

**Figure 1 vaccines-10-01185-f001:**
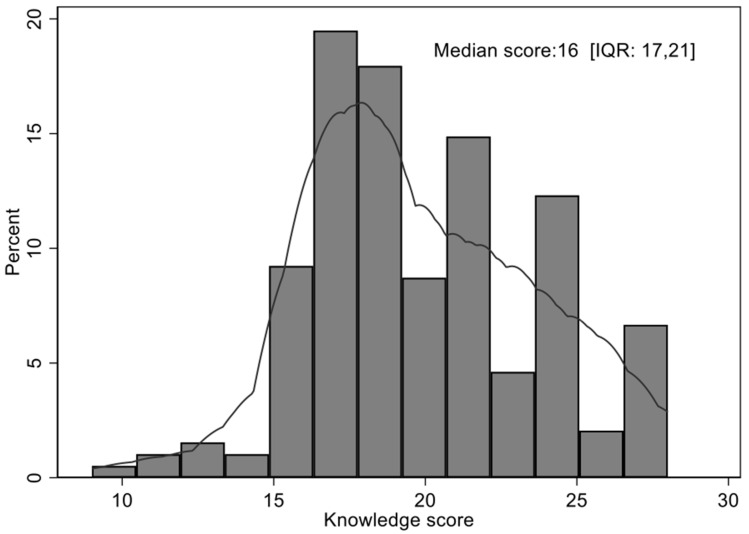
Histogram on the knowledge score of parents of adolescent children with regard to HPV infection and vaccination.

**Figure 2 vaccines-10-01185-f002:**
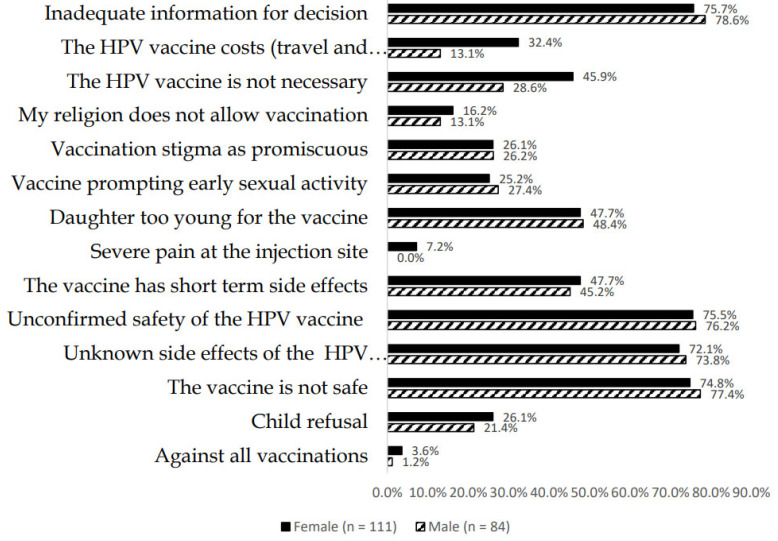
Reasons for HPV vaccination hesitancy.

**Table 1 vaccines-10-01185-t001:** Baseline sociodemographic characteristics of parents with adolescent children attending the medical clinic of Kenyatta National Hospital.

Participants (*n* = 195)		
	Frequency	Percentage (%)
**Age distribution (years)**		
18–30	12	6.2
31–40	79	40.5
>40	104	53.3
**Sex**		
Male	84	43.1
Female	111	56.9
**Occupation**		
Formal employment	54	27.7
Self-employment	122	62.6
Other	19	9.7
**Education Level**		
No formal education	1	0.5
Primary level	26	13.3
Secondary level	87	44.6
Tertiary	81	41.5
**Marital status**		
Married	165	84.6
Singlehood *	30	15.4
**Religion**		
Christian	192	98.5
Muslim	2	1
Other	1	0.5
**Age of children**		
9–11 years	70	35.9
12–14 years	125	64.1

NB: * Divorced/never married/widow/widower.

**Table 2 vaccines-10-01185-t002:** Parental knowledge of HPV vaccine.

Knowledge of HPV Vaccine (Proportion Responding YES or Correctly)	Males N (%)	Females N (%)	*p*-Value
Are you aware that all girls aged 10 years are being offered a human papilloma virus (HPV) vaccine?	61 (72.6%)	92 (82.9%)	0.062
What is the HPV vaccine used for?			
1. Prevention of HPV infection	45 (53.6%)	72 (64.9%)	0.222
2. Prevention of cervical cancer	50 (59.5%)	79 (71.2%)	0.04
3. Prevention of genital warts	30 (35.7%)	32 (28.8%)	0.563
What is the age group eligible for the HPV vaccine 9–26 years?	48 (57.1%)	84 (75.5%)	0.021
There is no need for Pap smear screening after receiving HPV vaccination	24 (28.6%)	66 (59.5%)	<0.001

**Table 3 vaccines-10-01185-t003:** Determinants of knowledge of HPV infection, cervical cancer and HPV vaccination.

	Bivariate Analysis		Multivariate Analysis	
	Crude Beta Co-Efficient (95% CI)	*p*-Value	Adjusted Beta Co-Efficient (95% CI)	*p*-Value
**Age** 18–30 (1) 31–40 (2) Above 40 (3)	−0.035 (0.845, −0.915)	0.937	−0.881 (−0.881, −2.609)	0.127
**Sex** Male (1) Female (0)	0.849 (1.927, −0.229)	0.124	−3.994 (−8.555, 0.566)	0.088
**Occupation** Formal employment (1) Self-employed (2) Other (3)	0.321 (1.238, −0.596)	0.493	-	-
**Education** Have no formal education (1) Primary level (2) Secondary level (3) Tertiary (4)	−0.71 (0.047, −1.467)	0.067	-	-
**Marital status** Married (0) Single (divorced/never married/widow/widower) (1)	0.121 (1.609, 1.786)	0.873	- -	-
**Religion**	1.114 (6.441, −4.213)	0.682	- -	-
**Sex of the children**	0.095 (0.775, −0.585)	0.783	- -	-

**Table 4 vaccines-10-01185-t004:** Generalized additive regression analysis for the determinants of the belief score.

Variables	Bivariate Analysis		Multivariate Analysis	
	Crude Beta Coefficient (95%CI)	*p* Value	Adjusted Beta Coefficient (95%CI)	*p* Value
**Age** 18–30 (1) 31–40 (2) Above 40 (3)	−0.057 (−0.305, 0.191)	0.652	-	-
**Sex** Male (1) Female (0)	0.192 (−0.154, 0.538)	0.278	0.255 (−0.056, 0.565)	0.11
**Occupation** Formal employment (1) Self-employed (2) Other (3)	0.273 (0.003, 0.544)	**0.049**	0.256 (−0.001, 0.512)	**0.052**
**Education** No formal education (1) Primary level (2) Secondary level (3) Tertiary (4)	0.075 (−0.157, 0.308)	0.526	-	-
**Marital status** Married (0) Single (divorced/never married/widow/widower) (1)	−0.182 (−0.631, 0.268)	0.43	-	-
**Religion** Christian (1) Muslim (2) Others (3)	0.681 (−0.818, 2.181)	0.374	-	-
**Adequate knowledge on HPV ***	0.057 (0.035, 0.080)	**<0.001**	−0.06 (0.039, 0.081)	**<0.001**

NB: * Knowledge scores of ≥15 out of 30 was considered as adequate knowledge.

**Table 5 vaccines-10-01185-t005:** Determinants of willingness to vaccinate child against HPV.

Variable	Crude OR		Adjusted OR	
	OR (95% CI)	*p*-Values	OR (95% CI)	*p*-Values
**Age** 18–30 (1) 31–40 (2) Above 40 (3)	0.512 (0.252, 4.043)	0.065	0.431 (0.194, 0.961)	**0.040**
**Sex** Male (1) Female (0)	0.96 (0.451,2.078)	0.933	2.369 (0.911, 7.643)	0.074
**Occupation** Formal employment (1) Self-employed (2) Other (3)	0.922 (0.482, 1.762)	0.806	-	-
**Level of education** Have no formal education (1) Primary level (2) Secondary level (3) Tertiary (4)	0.663 (0.310,1.419)	0.290	0.392 (0.188, 0.818)	**0.013**
**Marital status** Married (0) Single (divorced/never married/widow/widower) (1)	0.587 (.227, 1.512)	0.270	-	-
**Knowledge total score (%)**	0.969 (0.951, 0.988)	0.002	1.133 (1.050, 1.222)	**<0.001**
**Belief total score**	2.673 (1.859, 3.845)	<0.001	2.395 (1.604, 3.577)	**0.001**

## Data Availability

Additional data can be obtained on reasonable request from the corresponding authors.

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
