# Peer review of "Impact of Parental Knowledge and Beliefs on HPV Vaccine Hesitancy in Kenya—Findings and Implications"

_vaccines, 2022, doi:10.3390/vaccines10081185_

Round 1
Reviewer 1 Report
This descriptive cross-sectional study was conducted among 195 participants with children aged 9-14 years in Kenya. The authors found that despite the low levels of knowledge and safety concerns, there was a high parental willingness to have their children vaccinated. Furthermore, they also found the key reason for vaccine hesitancy was insufficient knowledge and safety concerns. This study gives some new insights into remaindering people to understand the important role of the HPV vaccine and collected valuable data in Kenya. However, some issues should be revised before publication.
1. Study design, location, and population: The participants in this study were enrolled only in a hospital (Kenyatta National Hospital), and the authors claimed that “KNH was selected for this study as it is a leading public tertiary care hospital in the Capital City of Kenya, Nairobi, representing an urban population.” I think this selection makes bias on the data collected from these participants because other populations such as people living in rural areas and suburbs were not included. It has been well known that income, education level, and medical care level have an impact on vaccine hesitancy and vaccination rate. Therefore, the results were underrepresented and unreliable.
2. Sample size determination: Although authors have claimed that the sample size was determined by using the Cochran formula, as a questionnaire-based study, I think the sample size is still too small to present the real situation.
3. According to the questionnaire, the knowledge score was determined based on Yes =1, No = 0, and I don’t know=2. For example, to the question ‘What is the HPV vaccine used for? (Yes =1, No = 0, I don’t know=2)”, the sum score of a participant with all correct answers (Yes) is 3, the sum score of a participant with knows nothing about this knowledge is 6, and the sum score of a participant with wrong answers is 0. In this case, what is the relationship between score and results? The authors claimed that the maximum knowledge score was 30, and the score was converted to a dichotomous variable using a cut-off value of greater than 15. Authors should clarify what is greater than 15 and what is less than 15.
4. Table 1: The baseline data in Table 1 should be statistically analyzed to identify potential confounders.
5. There is an error in the secondary title number, for example, Line 272 “3.2 Overall knowledge score”, and line 287 “3.2 Sources of information and knowledge of HPV infection, cervical cancer, and its prevention”. There are two 3.2 in the manuscript.
6. The contents in Lines 299-303 are repeated, please revise them.
7. Tables 3, 4, and 5 should be made into three-line tables.
8. The bar chart shown in Figure 2 is so narrow that the numbers on the right overlap.
9. The length of the Discussion is too long, please reduce it.
10. The language should be improved by an English-speaking person. For example, Title: “Impact of Parental Knowledge and Beliefs on Hpv Vaccine Hesitancy in Kenya; Findings and Implications” → “Impact of Parental Knowledge and Beliefs on HPV Vaccine Hesitancy in Kenya: Findings and Implications”. Line-19, “…vaccinations” → “vaccination”.
Author Response
This descriptive cross-sectional study was conducted among 195 participants with children aged 9-14 years in Kenya. The authors found that despite the low levels of knowledge and safety concerns, there was a high parental willingness to have their children vaccinated. Furthermore, they also found the key reason for vaccine hesitancy was insufficient knowledge and safety concerns. This study gives some new insights into remaindering people to understand the important role of the HPV vaccine and collected valuable data in Kenya. However, some issues should be revised before publication.
Author comment: Thank you for this review. We hope we adequately address your concerns.
- Study design, location, and population: The participants in this study were enrolled only in a hospital (Kenyatta National Hospital), and the authors claimed that “KNH was selected for this study as it is a leading public tertiary care hospital in the Capital City of Kenya, Nairobi, representing an urban population.” I think this selection makes bias on the data collected from these participants because other populations such as people living in rural areas and suburbs were not included. It has been well known that income, education level, and medical care level have an impact on vaccine hesitancy and vaccination rate. Therefore,the results were underrepresented and unreliable.
Author comments: Thank you for this comment. We have pointed out in the Introduction and Methodology why we necessarily adopted this approach for this comprehensive study involving both parents. We have also pointed out ion the limitations section areas of concern including the lack of rural parents. However, as mentioned, despite these concerns we believe our findings are robust adding to potential ways forward among key stakeholder groups to improve future vaccination rates in Kenya. We hope you agree.
Regarding whether the research design appropriate (Box comments)? We chose our particular study design as the alternative design would be longitudinal (cohort study or randomised control study). We believe longitudinal designs are in appropriate in this case because longitudinal studies seek to measure changes over time. In this case, we were interested in studying reasons for vaccine hesitancy at a given point in time. Hence, we believe a cross sectional design was appropriate. The cross-sectional study could have either been analytic or descriptive. We did not select an analytic design since we were not testing a hypothesis and the objectives of the study were of a descriptive nature. We also had a choice between a qualitative and quantitative descriptive cross sectional study. We selected a quantitative study because the methodological approach is standardised and the data is easier to analyse. In conclusion we still believe that the study design we selected was most appropriate, and hope you agree with us
Regarding whether the conclusions supported by the results (Box comments), we have deleted the last two sentences in the conclusion since they were not very relevant.
- Sample size determination: Although authors have claimed that the sample size was determined by using the Cochran formula, as a questionnaire-based study, I think the sample size is still too small to present the real situation.
Author comments: Thank you for this comment. We agree that in epidemiological studies, the larger the size of the sample the greater the precision of the findings and external validity. However, its well-recognised in epidemiology that increases in sample size are associated with increased costs, and one has to strike a balance between feasibility and a large sample size. We believe that given the Cochran formular which is widely accepted, this was the optimal approach for the sample size computation. There can also be pitfalls with very large sample sizes, with a new reference now added in regarding criticisms about inflating the sample size. We hope this is now acceptable
- According to the questionnaire, the knowledge score was determined based on Yes =1, No = 0, and I don’t know=2. For example, to the question ‘What is the HPV vaccine used for? (Yes =1, No = 0, I don’t know=2)”, the sum score of a participant with all correct answers (Yes) is 3, the sum score of a participant with knows nothing about this knowledge is 6, and the sum score of a participant with wrong answers is 0. In this case, what is the relationship between score and results? The authors claimed that the maximum knowledge score was 30, and the score was converted to a dichotomous variable using a cut-off value of greater than 15. Authors should clarify what is greater than 15 and what is less than 15.
Author comments: Thank you for this comment. We realise that we did not add in our final conversion of ‘No’ and ‘Do not know’ to zero and yes to One to help with the various analyses. We have now done so and also clarified what the knowledge scores of less or greater than 15 mean. We hope this is now OK.
- Table 1: The baseline data in Table 1 should be statistically analyzed to identify potential confounders.
Author comments: Thank you for this question. However, the study design was not analytic. In analytical study designs that have two or more groups, we would have performed inferential tests such as chi square to identify significant differences across arms. However, in a single arm study this is not typically undertaken. We hope this is acceptable to you.
- There is an error in the secondary title number, for example, Line 272 “3.2 Overall knowledge score”, and line 287 “3.2 Sources of information and knowledge of HPV infection, cervical cancer, and its prevention”. There are two 3.2 in the manuscript.
Author comments: This error is noted than you -and corrected
- The contents in Lines 299-303 are repeated, please revise them.
Author comments: Thank you - now addressed.
- Tables 3, 4, and 5 should be made into three-line tables.
Author comments: This is noted and corrected
- The bar chart shown in Figure 2 is so narrow that the numbers on the right overlap.
Author comments: This is noted and corrected
- The length of the Discussion is too long, please reduce it.
Author comments: Thank you for this comment. We have now appreciably reduced the length of the Discussion to concentrate on the key findings and their implications. We hope this is now acceptable.
- The language should be improved by an English-speaking person. For example, Title: “Impact of Parental Knowledge and Beliefs on Hpv Vaccine Hesitancy in Kenya; Findings and Implications” → “Impact of Parental Knowledge and Beliefs on HPV Vaccine Hesitancy in Kenya: Findings and Implications”. Line-19, “…vaccinations” → “vaccination”.
Author comments: Thank you now done. One of the co-authors is a native English speaker with over 450 publications in peer-reviewed Journals to his name since 2008, and has revised the paper to improve the English and the flow. We hope this is now OK.
Reviewer 2 Report
- Title: please correct to “HPV” (all capital letters)
- Abstract, line 2: Please rephrase the following sentence: Consequently, a need to identify 20 prevalence and risk factors for HPV vaccination hesitancy among parents in Kenya.
- Introduction, line 43: please replace “believed” with “estimated”.
- Methods, line 173: “One some days”, please correct.
- Methods, line 202: please change “Hard copies of the raw data were stored under 202 lock and key, and only the principal investigator had access to the documents” to “only the principal investigator had access to the documents”.
- Results: In line 260 you mention “Other than two participants, all were 260 Christians”, however in Table you present three non-Christian persons. Please correct accordingly.
- Please improve Figure 2.
- Do we know the vaccination coverage against HPV in Kenya?
- Beyond parents, healthcare personnel also have an impact on vaccine decisions. Therefore, you may want to use the following article in the reference list:
Papillomavirus Res 2019 Dec;8:100183. doi: 10.1016/j.pvr.2019.100183. Epub 2019 Aug 30.
The role of healthcare providers in HPV vaccination programs - A meeting report
Author Response
- Title: please correct to “HPV” (all capital letters)
Author comments: Thank you – now addressed
- Abstract, line 2: Please rephrase the following sentence: Consequently, a need to identify 20 prevalence and risk factors for HPV vaccination hesitancy among parents in Kenya.
Author comments: Thank you now addressed
- Introduction, line 43: please replace “believed” with “estimated”.
Author comments: Thank you – now addressed
- Methods, line 173: “One some days”, please correct.
Author comments: Thank you – now addressed
- Methods, line 202: please change “Hard copies of the raw data were stored under 202 lock and key, and only the principal investigator had access to the documents” to “only the principal investigator had access to the documents”.
Author comments: Thank you – now addressed.
- Results: In line 260 you mention “Other than two participants, all were 260 Christians”, however in Table you present three non-Christian persons. Please correct accordingly.
Author comments: Thank you - the error has been noted and changes made.
- Please improve Figure 2.
Author comments: Thank you – Figure 2 has been improved. We hope this is now acceptable.
- Do we know the vaccination coverage against HPV in Kenya?
Author comments: Thank you for this – comments have now been added in to the Introduction, etc. We hope this is now OK.
- Beyond parents, healthcare personnel also have an impact on vaccine decisions. Therefore, you may want to use the following article in the reference list: Papillomavirus Res 2019 Dec;8:100183. doi: 10.1016/j.pvr.2019.100183. Epub 2019 Aug 30.
The role of healthcare providers in HPV vaccination programs - A meeting report
Alex Vorsters 1, Paolo Bonanni 2, Helena C Maltezou 3, Joanne Yarwood 4, Noel T Brewer 5, F Xavier Bosch 6, Sharon Hanley 7, Ross Cameron 8, Eduardo L Franco 9, Marc Arbyn 10, Nubia Muñoz 11, Mira Kojouharova 12, Jade Pattyn 13, Marc Baay 14, Emilie Karafillakis 15, Pierre Van Damme 13
Author comments: Thank you – now added in.
Reviewer 3 Report
The author makes a lot of effort to research the HPV vaccination from low- and middle-income nations (LMICs) in the current article by Chester et al., on Impact of Parental Knowledge and Beliefs on HPV Vaccine Hesitancy in Kenya. In order to inform future strategies, the author identifies the factors that cause vaccine hesitation among Kenyan parents that visit clinics and conclude that despite the lack of understanding and safety worries, parents were very keen to get their kids immunized. Insufficient information and safety worries were the main causes of vaccine reluctance. Compared to women, men typically had less understanding about the HPV vaccine. Most parents would respect their kids' decisions. Since it is well written and beneficial for the HPV vaccination program, no additional significant revisions are necessary, yet slight adjustments might improve the content.
- An abstract must be written with clarity in mind.
- In the methods section, the author must make it clear how many individuals were included in the study (sampling and participant recruitment).
- In lines 448-449, the author stated that men had less decision-making knowledge about the HPV vaccine than did women, yet at the same time, lines 544-545 stated that younger women knew less about HPV than did men in the same age range. To support a claim like a comment, solid justification and evidence are needed.
- The author has to talk a little bit more about how the future will see an improvement in the poor relationship between parental education and vaccination willingness.
- If at all possible, the author should also choose a random volunteer from a remote location.
- It would be wonderful if the author could include some participants from places other than hospitals, such from public spaces.
- The study's statistical power has to be discussed.
- How common is HPV vaccination in Kenya today?
- Has the author approached any other hospitals about conducting a similar study? Or have planned in upcoming future?
Author Response
The author makes a lot of effort to research the HPV vaccination from low- and middle-income nations (LMICs) in the current article by Chester et al., on Impact of Parental Knowledge and Beliefs on HPV Vaccine Hesitancy in Kenya. In order to inform future strategies, the author identifies the factors that cause vaccine hesitation among Kenyan parents that visit clinics and conclude that despite the lack of understanding and safety worries, parents were very keen to get their kids immunized. Insufficient information and safety worries were the main causes of vaccine reluctance. Compared to women, men typically had less understanding about the HPV vaccine. Most parents would respect their kids' decisions. Since it is well written and beneficial for the HPV vaccination program, no additional significant revisions are necessary, yet slight adjustments might improve the content.
Author comments: Thank you for this summary. We hope we adequately address your concerns/ issues.
- An abstract must be written with clarity in mind.
Author comments: Thank you – now addressed where we can
- In the methods section, the author must make it clear how many individuals were included in the study (sampling and participant recruitment).
Author comments: Thank you – this information is documented at the start of the Results section (3.1). We hope this is now OK.
- In lines 448-449, the author stated that men had less decision-making knowledge about the HPV vaccine than did women, yet at the same time, lines 544-545 stated that younger women knew less about HPV than did men in the same age range. To support a claim like a comment, solid justification and evidence are needed.
Author comments: Thank you - This effect that we observed was effect measure modification. We did not provide evidence for these in the methodology details because in our considerable experience over multiple publications such as these - many readers just do not understand the concept of effect measure modification. In effect measure modification - the relationship between two variables changes according to the values of a third variable known as the effect measure modifier. We studied this interaction effect in detail but placed the findings in the supplementary material for those interested. Please see Figure D: The effects of Age and Gender on Knowledge Score. We hope this is now OK.
- The author has to talk a little bit more about how the future will see an improvement in the poor relationship between parental education and vaccination willingness.
Author comments: Thank you – now added in more comments in the Discussion based on comments made in the Introduction and Results. We hope this is now acceptable.
- If at all possible, the author should also choose a random volunteer from a remote location.
Author comments: Thank you for this comment. We have now expanded in the Methodology why we chose KNH for this study. In order to make meaningful comments regarding any answers from participants in remote areas – we would need to include an appreciable number of suitable rural parents, which was just not practical on this occasion. We hope this is now clear.
- It would be wonderful if the author could include some participants from places other than hospitals, such from public spaces.
Author comments: As stated above - we have now expanded in the Methodology why we chose KNH for the study – particularly during a pandemic. We hope this is also now clear - although we hope to expand this study.
- The study's statistical power has to be discussed.
Author comments: Thank you for this. As mentioned, this study was not hypothesis driven. However, on regression modelling we did make some statistically significant findings at a level of alpha 0.05. We hope this is now clear.
- How common is HPV vaccination in Kenya today?
Author comments: Thank you. We have now provided additional information on vaccine coverage in Kenya in 2020 with the appropriate reference. We hope this is now OK.
- Has the author approached any other hospitals about conducting a similar study? Or have planned in upcoming future?
Author comments: Not yet, but we are planning to so as well as rural areas.
Round 2
Reviewer 1 Report
The authors have revised the manuscript following my comments, I would like to give acceptance to this submission.